# Assessment Possibilities of the Quality of Mining Equipment and of the Parts Subjected to Intense Wear

Vlad Alexandru Florea [1,*] , Mihaela Toderaş [2] and Răzvan-Bogdan Itu [1]

[1] Department of Mechanical, Industrial and Transportation Engineering, University of Petrosani, 332006 Petrosani, Romania
[2] Mining Engineering, Surveying and Civil Engineering Department, Faculty of Mines, University of Petrosani, 332006 Petrosani, Romania
* Correspondence: vladflorea@upet.ro; Tel.: +40-7-2234-8000

**Abstract:** The equipment in underground mines provides a continuous production flow, depending on the way their quality is preserved during their operation. The TR-7A scraper conveyer subassemblies, which function in the Jiu Valley coal basin and are subjected to abrasion wear, showed a high failure frequency (chains, chain elevators, and driving and turning drums), as well as the hydraulic couplings and certain electric equipment of the same machinery. The data collected following the TR-7A scraper conveyer at work allowed the parameters to be determined that characterise the reliability and maintainability of the above-mentioned components, the failure modes, and their effects. Using calculation methods, the interpretation of the results has been facilitated, with a view to reducing maintenance costs and obtaining an 80% reliability for the components with the most failures, in the case of the TR-7A scraper conveyer.

**Keywords:** conveyer; wear; maintenance; reliability; maintainability; cause–effect diagram

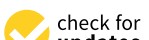



## 1. Introduction

For all organizations and companies, a major disadvantage in terms of productivity and production results is encountering various production equipment failures, which automatically lead to lost production time and, implicitly, to increased expenses at the organization level. In this sense, the development of a maintenance strategy for production equipment is of primary interest. More and more companies worldwide are implementing preventive maintenance systems, which is a proactive maintenance strategy that involves the regular and routine maintenance planning of production equipment. By regularly scheduling the cleaning, repairs, adjustments, and replacement of component parts, a preventive approach increases the reliability over time and the productivity of work equipment. For companies that have not implemented a preventive maintenance procedure, the costs of operating production equipment can exceed up to 10 times the costs of a company that has such a well-established preventive maintenance procedure.

The advantages of preventive maintenance include a significant increase in the level of safety of work equipment and, thus, a drastic reduction in the risks of accidents, the reduction of losses in the production process, fewer unexpected breakdowns of essential equipment for production, and reduced costs regarding possible emergency repairs. However, preventive maintenance also involves a number of disadvantages, including the possibility of excessive preventive maintenance, high implementation costs, and the need for more resources in terms of time and personnel. Even in this case, weighing the advantages and disadvantages of implementing a preventive maintenance system, it can be concluded that, in terms of development and a long-term, safe operation, it can bring very important benefits. A vitally important benefit that preventive maintenance can bring to an organisation is that it helps to anticipate faults and intervene to remedy them before they

reach the critical point when the handling and operation of the equipment could represent a potential danger to the operator.

A particularly important problem in quality assessment is keeping the quality up for as long as possible. The complexity of production systems and the optimization of maintenance systems have become a topic that has begun to attract the attention of many researchers [1–8]. In mining operations, the role of maintenance is to optimise the speed, performance, and quality of the technological operations that are carried out in order to extract useful mineral substances. The well-performed maintenance of any mining equipment leads to the achievement of a minimum cost of their life cycle, to the reduction of the long-term costs of the mining companies, but, above all, to the reduction of the dangers that could arise due to the failure of the equipment, especially in the case of coal mines [8]. As shown by Dhillon [9], the current requirements of the economy require the creation of products used in engineering that are very reliable and easy to maintain. In the studies conducted by Ruschel et al. [8], a series of information is presented on how to make decisions regarding industrial maintenance, the proposed models, and the application of methods and tools.

In the field of engineering, as with any production system, the most important elements are the production, maintenance, and quality of the operations and the products obtained. Any production equipment must be designed to work for a long period of time [10–15]. However, their performance is affected by progressive degradation or wear until failure occurs.

The mining industry was and remains one of the most challenging work sectors. The mining sector is the main source of national wealth for many countries. The mining field is a competitive one, and the equipment used in this field of activity is subject to difficult conditions and works in dangerous environments. Mining operations depend on the efficient use of mining equipment. Any technological mining operation is focused on production costs and on obtaining and maintaining a profit. In order to have productive mining and operations that are carried out as easily as possible with maximum security and safety, it is necessary to have mining equipment that is as reliable as possible. Reliability in the mining industry is a particularly important issue because mining equipment operates in dangerous working conditions, whether it is work carried out in quarries or, especially, in the case of underground exploitation. The mining equipment must present safety and efficiency and involve costs as low as possible. Maintenance of this equipment is more efficient in terms of costs than replacing the entire equipment or some of its components, which would be too expensive [16,17].

Maintenance in safe conditions is in the interest of employers. The appropriate management of safety and health at work is beneficial to society and a characteristic of efficient organisations. Through maintenance, a link is established between the appropriate management of safety and health at work and quality assurance procedures. The lack of equipment maintenance or their improper maintenance can lead to some particularly dangerous situations, work accidents, and even health problems [18–23].

Maintenance operations are divided into two categories, namely, repair operations and maintenance operations [24–26]. As maintenance systems, they are divided as follows:

- Preventive maintenance (systematic, conditional, and predictive) represented by all the systematic interventions, performed at regular time intervals, which aims to ensure the correct functioning of the systems;
- Corrective maintenance (curative and palliative), representing interventions as a result of minor, accidental breakdowns, which aims to restore the product's functioning capacity.

In order to maintain the quality of production equipment, it is important that it is always in a condition similar to that of the equipment at the time it was put into operation. Therefore, the equipment must be checked and subjected to planned maintenance, which implies a close connection between production planning and that of preventive maintenance, so that the downtime of the equipment is as low as possible [23–27].

High-quality mining machinery should be reliable in operation, keeping their initial performances for as long as possible. In this sense, the primordial role is played by availability, which, in turn, is related to reliability and maintainability. The science of quality evolution, by indicators of machinery reliability and maintainability, involves monitoring those at work, noting the faults occurring in the functioning and remedy time, as well as their interpretation [16–21]. It is important that the mining equipment is of superior quality, which is particularly important in the case of technological operations carried out underground at different depths and in difficult geological and mining conditions.

The conditions referring to the quality system, applied based on SR EN ISO series 9000 standards, can be called "Model for quality assurance in design, development, production, assembly and service", which includes all the stages of creation of a product or service, from design and execution to the use by the beneficiary; one of the stages refer to corrective and preventive actions that would be included in the quality management system that the enterprises decide to apply [28].

The rapid development of science and technology and the restructuring of the extractive mining industry determine the profound changes in the structure and complexity of the machinery used in mining. Mines should be equipped with machines and installations of high technicity, which would provide an advanced degree of mechanization and automation, thus ensuring increased productivity.

In the conventional view, the unique quality co-ordinate is represented by functionality, the capacity of a product to fulfill the function for which it had been created. In the mining machine, machinery, and equipment industry, this functionality expresses, as a rule, those quality elements that correspond to the requirements, standards, technical documentations, or other normative documents [29–31].

The intensification of technological processes, increase of working speed, special operating conditions, and high degree of strain imply, from a technical and economic point of view, an increased responsibility in providing the functionality of mining machinery, machines, and installations [32].

Modern technologies and their high degree of complexity, thus, impose the improvement of activities, both in design and in manufacturing, and in maintenance and repair. The wear parts of mining equipment are commonly replaced components [32].

Although studies have been performed regarding the quality of mining machinery, those were very few and with rather scarce data collected. This is due to several factors:

- A great variety of mining machine and machinery types, on the one hand, and to a larger extent, different operating conditions. It is quite difficult to appreciate the quality of mining machinery, since, in order to do this, several observations should be processed and interpreted;
- The lack of an informational system, of recording and keeping information of the failures per mining machinery type. Moreover, some of the endeavours of collecting data referring to the quality of the machinery are made difficult by those who are in charge in this sense, due to not knowing the specific problems, the importance of these studies, and the methods of analysis;
- Several causes that lead to failures of the mining machinery, which might include the exploitation mode, which make the establishing of the type of fault repair function difficult;
- The lack of references regarding the quality of the mining, which is the cause of why certain methods applied and the results obtained might not be verified and compared. In this sense, other traditional fields of expertise are chosen, which is an accumulated experience, the particularities of the mining machinery not being excluded.

The design, choice, execution, exploitation, maintenance, and repair of mining equipment are influenced by the general and specific conditions of their operation, namely that:

- The small assembling and transportation space imply smaller sizes;
- The inclined working position should not affect the greasing of the moving parts, and seals should not allow lubricant leakage;

- The atmosphere is dust-polluted and has high humidity;
- The working faces are in continuous movement, determining the machinery being moved without being disassembled (scraper conveyers at the face);
- The short-term overload takeover, the overload occurring due to flow, which can operate in variations and modifications in the conditions at the face.

## 2. Materials

Among the machinery that makes up underground flows, the TR 7A conveyer can be found (Figure 1), which can operate together with miners of up to 400 kN and be used in faces (with an inclination in the range of +35° and −35°) equipped with individual supports. The machinery operates in faces found on floors with a minimum admitted specific resistance to compression of 50 N/cm$^2$.

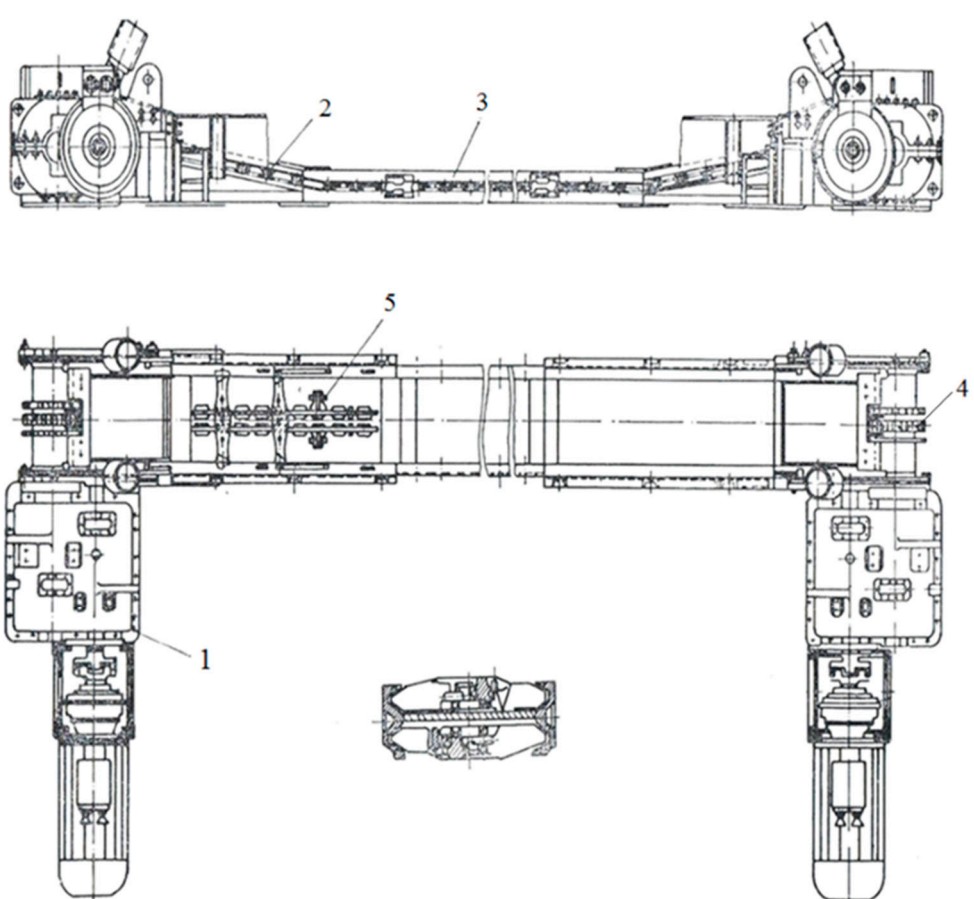

**Figure 1.** TR-7A scraper conveyer: 1—driving station made up of an electric motor, hydraulic gear, guriflex gear, and reduction gear; 2—intermediary trough; 3—trough column; 4—chain star; 5—conveyer scrapers and chains.

Table 1 shows the main technical characteristics of TR-7A scraper conveyer.

**Table 1.** Technical characteristics of TR-7A scraper conveyer.

| Technical Characteristic of TR-7A Scraper Conveyer [1] | UM | Value |
|---|---|---|
| Transport capacity | t/h | 450 |
| Transport length | m | 60; 120 |
| Chain speed | m/s | 0.7; 0.9 |
| Installed power | kW | $2 \times 125$ |
| Electric supply voltage | V | 660 |
| Number of chain branches | pcs | 2 |
| Scraper pace | mm | 920 |
| Trough height | mm | 220 |
| Total height | mm | 1210 |
| Total width | mm | 3914 |
| Total weight | t | 66 |

[1] TR-7A scraper conveyer from Lonea Mine—Jiu Valley coal basin.

Mechanical defects noticed during operation of the conveyer were:

- At the driving stations: leaks at the covers of the reduction gear housing bearings, hydraulic gears with no hydraulic oil, blocked by rigid elements, serious wear at the teeth of the driving star (in the areas of contact with the chain);
- Alongside with the conveyer: serious wear of chain, bent, out of shape or missing scrapers, as well as worn troughs;
- In the electric conveyer driving installation: blocking of switches and cables with damaged insulation, unsealed electric switchgear boxes.

All these defects are due to poor maintenance leading to: frequent breaking of the chain; burning out of the motor; oil loss in the reduction gear (or visiting); failure of strike pinion in stage I of the reduction gear; teeth breaking in the reduction gear due to overload; chain coming out of troughs and stopping of the conveyer; destroying the discharge plate at the driving station; breaking the electric cable; electric faults in the coupling–interruption switchgear boxes.

Certain works of specialty analyze production losses, most often statistically, due to failures, without indicating adequate solutions for solving the problems.

Although considerable improvements have been made, in the sense of increasing the operating time of the subassemblies and of the spare parts of the mining equipment, the equivalent increase of service time of the equipment of which they are a part did not correspondingly increase, so that there have been cases in which the latter's life span was shorter than that of their components. Consequently, an analysis of the "lifestyle" in time of the equipment and their component parts is required, in specific operating conditions, in order to determine ways of increasing their reliability. Currently, it is necessary to find materials and methods for reconditioning spare parts and subassemblies that most frequently become damaged, in view of reducing production costs in coal mines.

The principal physical–mechanical characteristics that the respective materials of which the subassemblies for the mining equipment are made should possess are: high mechanical resistance (hardness associated with tenacity), alongside with resistance to abrasion wear; these characteristics are usually in inverse proportional ratio.

The paper analyzes the failure times and remedy times in the case of the benchmarks of the TR-7A scraper conveyer, with the intention of making TR-7A conveyer maintenance efficient.

### 3. Methods

The methodology on which the reliability and maintainability analysis was based has the following stages [17]:

1. Identification and analysis of factors influencing the wear of the TR-7A conveyer components by using a cause–effect diagram;

2.   Analysis of the reliability and maintainability indicators with a view to determining solutions for making maintenance efficient.

*3.1. Cause–Effect Diagram (Ishikawa)*

An Ishikawa diagram (Figure 2) is a type of analysis that uses a graphological, causal, and suggestive method with the aim of optimising the quality of a product, allowing efficient and economic action on the technological parameters of the process. The first cause-and-effect diagram was built by Kaoru Ishikawa from the University of Tokyo in 1953, which is why it is called the Ishikawa diagram [33–35]. Similarly, it is also known as a "fishbone diagram, due to its graphic representation resembling a fish skeleton.

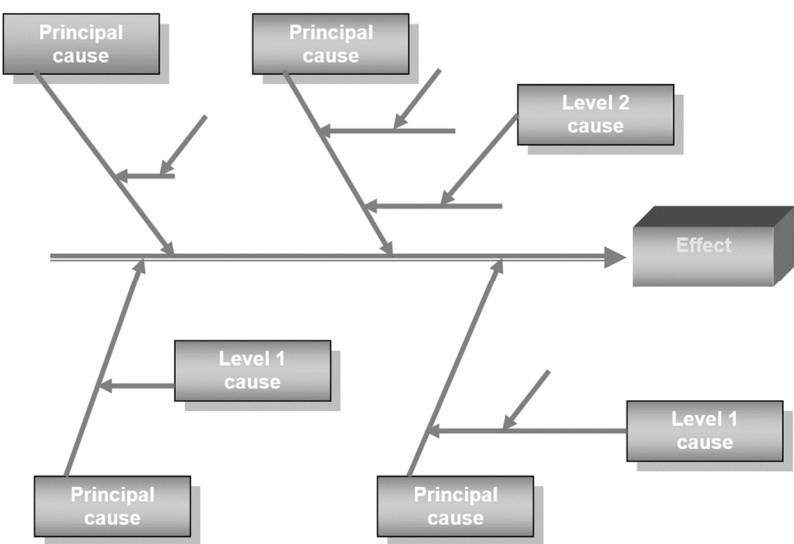

**Figure 2.** Cause–effect diagram (Ishikawa).

The proposed methodology for the building of the diagrams structured on the components of the process implies the following basic stages to be covered:

1.   Establishing quality characteristics that are wished to be improved or kept under control;
2.   The quality characteristic is written on the right side of the format in which we wish to build the diagram, and it is framed in a rectangle;
3.   The principal line (arrow) is drawn from the left side to the right; the arrow can be represented by a double line.
4.   Establishing the principal causes influencing the quality characteristic. The principal causes are also framed in rectangles. From these causes, arrows are directed that make ramifications and branches. It is recommended to group as great a number as possible of possible causes of the noticed dispersion (represented as branches). On each of these branches, in the stages that follow, the causes and factors determining them are described in greater detail, forming branchlets. Identification of the variation causes of the quality characteristic dispersion can be more easily made if successive questions are asked and answered, including "Why?" and "How is dispersion influenced by these groups of causes?" If the diagram is made by the members of a circle of quality, for individualization of causes, brainstorming can be used. The possible causes of dispersion of the quality characteristic should thus be pointed out in the diagram, so that every mutual relation would be clear;
5.   Establishing secondary causes that influence the principal ones based on the analysis of the quality characteristic variation (function of the principal causes) and are represented as medium ramifications;

6.  Establishing tertiary causes that influence secondary causes based on the analysis of the quality characteristic variation and function of the secondary causes and are represented by small branches;
7.  Establishing factors influencing tertiary causes;
8.  Verification of the inclusion of all the possible causes of the quality characteristic dispersion;
9.  Marking the important particular causes, which have a significant effect on the quality characteristic.

The domains of use for this diagram are practically unlimited, being able to be applied in solving any problems based on the in-depth study of cause–effect interaction. Cause–effect diagram can be developed for:

-   Identify the causes of the increase of the number of failures within a technological process;
-   Discover the causes of failure to achieve quality of conformity;
-   Establish the causes of the increase of the number of defects in machinery;
-   Establish the causes of decrease of productivity;
-   Establish the causes of decrease of sales and profit margins;
-   Establish the causes of occurrence of unsalable stocks;
-   Establish the causes of increase of complaints, with negative impact on the image of the company.

*3.2. Fundamental Indicators of Reliability and Maintainability*

Fundamental indicators of reliability, as magnitudes that quantitatively express reliability, are expressed by:

-   Probability of good operation—reliability function *R(t)*;
-   Probability of failure—unreliability function *F(t)*;
-   Failure probability density *f(t)*;
-   Failure intensity or rate $\lambda(t)$;
-   Mean time between failures *MTBF*;
-   Mean time to repair *MTR*;
-   Repair rate $\mu$.

From quantitative point of view, reliability was defined as being the probability for a product (technical system) to fulfill its mission (fundamental function), for a pre-established period of time, in certain given conditions [36–39].

According to this definition, the probability of good operation *p(t)*, that is reliability *R(t)*, is expressed by the equation:

$$p(t) = R(t) = P(t > t_i), \tag{1}$$

where *t*—random time variable (time of mission) and $t_i$—specified limit of the duration of good operation.

In an experimental determination, in order to obtain an analytical form of function, it is supposed that a behavior mode in time of a statistic population made up of $N_0$ new identical products is followed, which perform in the same work conditions and which have been manufactured based on the same technology, with the same imposed conditions. Considering that, at the moment *t* = 0, all the other $N_0$ products are in a state of operation, then at moment $t_i$, found in the range of [*t*, *t* + $\Delta$t], there are only *N* products in a state of operation. Thus, along $\Delta t$ time, it is considered that $\Delta N = N_0 - N$ flawed products.

The proportion of products in state of operation at moment $t_i$, that is, reliability $\hat{R}(t)$, is given by the ratio:

$$\hat{R}(t) = \frac{N}{N_0}, \tag{2}$$

The analytical form of the empirical function of reliability is expressed in this form:

$$\hat{R}(t_i) = \frac{N_{t_i}}{N_0},$$ (3)

where $N_{t_i}$—the number of products (elements) being in operation at moment $t_i$.

The non-reliability function expresses the failure probability of a product that should perform well for an established period of time $t_i$, under certain given conditions. It is expressed by the equation:

$$F(t) = P(t < t_i),$$ (4)

Between the reliability and non-reliability functions, the following relationship exists:

$$F(t) = 1 - R(t),$$ (5)

The analytical form of the empirical function of non-reliability is expressed in the form:

$$\hat{F}(t_i) = \frac{N_0 - N_{t_i}}{N_0},$$ (6)

By deriving the non-reliability function in time, a new function is obtained, called the density of probability of failures $f(t)$, which is the function of frequency or density of distribution and which expresses the relative frequency of failures in a time interval $dt$:

$$f(t) = \frac{dF(t)}{dt} = -\frac{dR(t)}{dt},$$ (7)

The analytical form of the empirical function which expresses density of probability is:

$$\hat{f}(t_i) = \frac{\Delta N_i}{N_0 \cdot \Delta t_i},$$ (8)

where $\Delta N_i = N_{t_{i-1}} - N_{t_i}$, $N_{t_i}$, and $N_{t_{i-1}}$—number of elements in operation at moment $t_i$, respectively; $t_{i-1}$; $\Delta t_i$—duration of the interval considered.

The proportionality factor $\lambda(t)$ represents one of the most important parameters of reliability and is called failure rate (failure rate or intensity of failure).

The formula $\lambda(t)dt$ is the probability that an equipment, in good condition at time $t$, would fail in the interval $(t, t + dt)$. By definition, $\lambda(t)dt$ is a density of conditioned failure probability. The failure rate or intensity is determined with the equation:

$$\lambda(t) = \frac{f(t)}{R(t)} = \frac{-\frac{dR(t)}{dt}}{R(t)},$$ (9)

By solving the differential Equation (9), taking into consideration the limit condition, the general expression of reliability is finally obtained:

$$R(t) = e^{-\int_0^t \lambda(t)dt},$$ (10)

The limit condition means that at moment $t = 0$, that is, at the moment in which the reliability study starts, the products are in a state of operation, which means that $R(0) = 1$. Experimental determination of the failure intensity for a time $\Delta t_i$, a function of the absolute frequency of failures in the time interval considered, is performed with the equation:

$$\hat{\lambda}(t_i) = \frac{\hat{f}(t_i)}{\hat{R}(t_i)} = \frac{\Delta N_i}{N_{t_i} \cdot \Delta t_i},$$ (11)

The average time of good operation expresses the average operation time until failure occurs, in case of un-reparable components or between two consecutive failures, in the case of reparable failures.

The value of the parameter is given by the average of operation times, which, for a continuous distribution, at limit, can be written as:

$$m = MTBF = \int_0^\infty R(t)dt = \int_0^\infty [1 - F(t)]dt = \int_0^\infty e^{-\int_0^t \lambda(t)dt}, \tag{12}$$

where *MTBF*—mean time between failures.

In the case of a batch of $N_0$ products, each of them has a certain duration of operation, $t_{f_i}$. The average of a good operating time is determined, in this case, based on the discrete values in effect, as the arithmetic average of a good operating time $t_{f_1}, t_{f_2}, \dots, t_{N_0}$:

$$\widehat{MTBF} = \frac{\sum_{i=1}^{N_0} t_{f_i}}{N_0}, \tag{13}$$

The mean time of repair *MTR* gives information regarding the number of hours referring to a reparation. Most often, it is expressed in hours/reparation and is calculated with the equation:

$$\widehat{MTR} = \frac{\sum_{i=1}^{n} t_i}{n}, \tag{14}$$

where $t_i$—time required to perform maintenance and *n*—total number of maintenance action.

Reparation rate represents the reparation intensity of a product, that is, the density of the conditioned probability of finishing a reparation in a timespan ($t, t + \Delta t$), starting from the hypothesis that the product was in repair in the interval (0, *t*). It is usually expressed in reparations/hour and is defined as being the inverse of the average repair time:

$$\hat{\mu} = \frac{1}{\widehat{MTR}}, \tag{15}$$

## 4. Results and Discussion

### 4.1. Maintenance Improvement of TR-7A Scraper Conveyer Using Cause–Effect Diagrams (Ishikawa)

Figures 3–6 show Ishikawa diagrams that point out the causes leading to the failure of the following elements of the TR-7A type conveyer: conveyer chain, hydraulic coupling, electric actuating installation, and the actuating motor (burnout).

The diagram in Figure 3 highlights that the interruptions in operation due to the chain have the following causes: operating personnel (negligent, unresponsive, and poorly trained); the conveyor has inadequate technical characteristics, especially gauge dimensions; materials and manufacturing technologies are inadequate (the material heats up by friction against the conveyor chute, has inclusions, and has a defective heat treatment); and the maintenance and repair methodology is inadequate (without adequate equipment).

The diagram in Figure 4 highlights that the interruptions in operation due to the coupling have the following causes: operating personnel who are careless, unresponsive, or poorly trained; the conveyer's technical characteristics, in particular, its insufficient transport capacity; the materials, especially the oil used in the coupling, are inadequate; and there is no appropriate equipment and a plan for revisions and repairs.

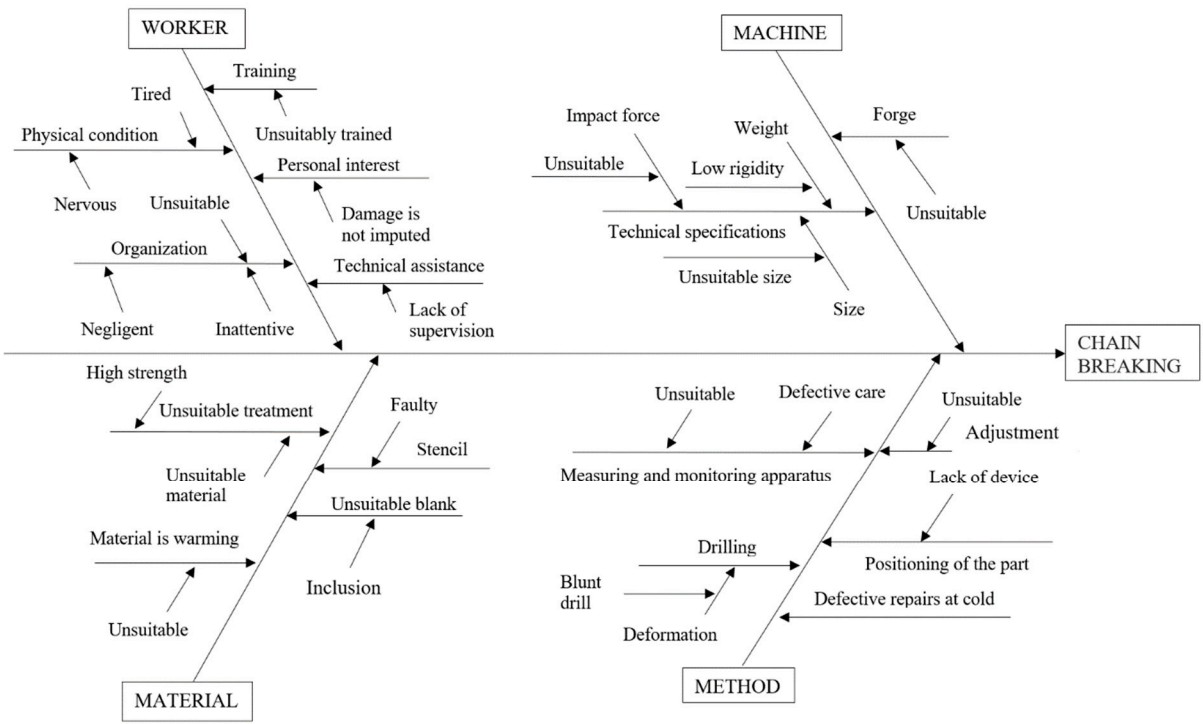

**Figure 3.** Ishikawa diagram (cause–effect) for chain break failure.

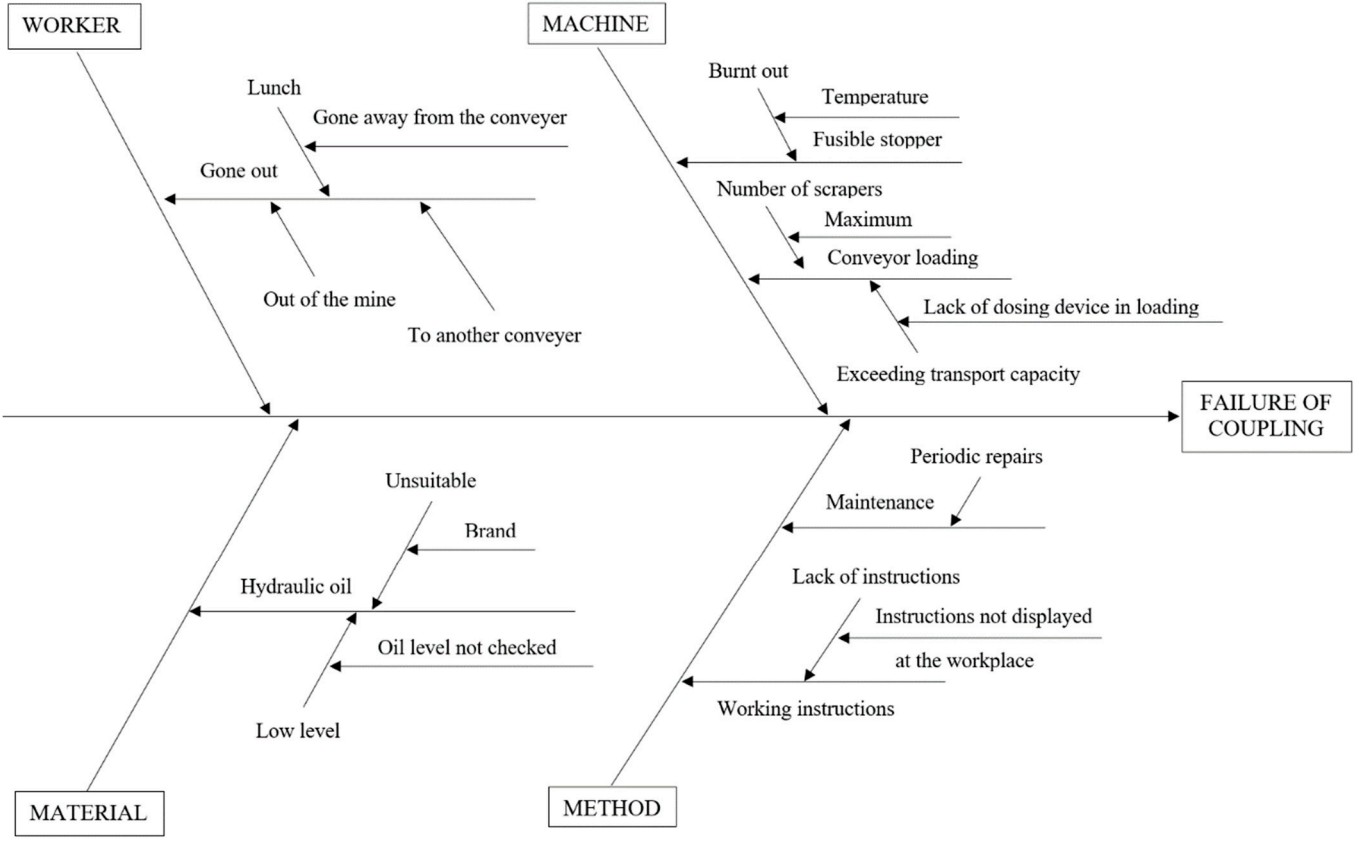

**Figure 4.** Ishikawa diagram (cause–effect) for coupling failure.

The diagram in Figure 5 highlights that the interruptions in operation due to the electrical components have the following causes: the measurement and monitoring of the control elements are defective; the operating staff is poorly trained; the conveyer has many interruptions in operation due to the operating conditions with humidity and dust; and the electrical materials and components (cables and switches) do not work properly.

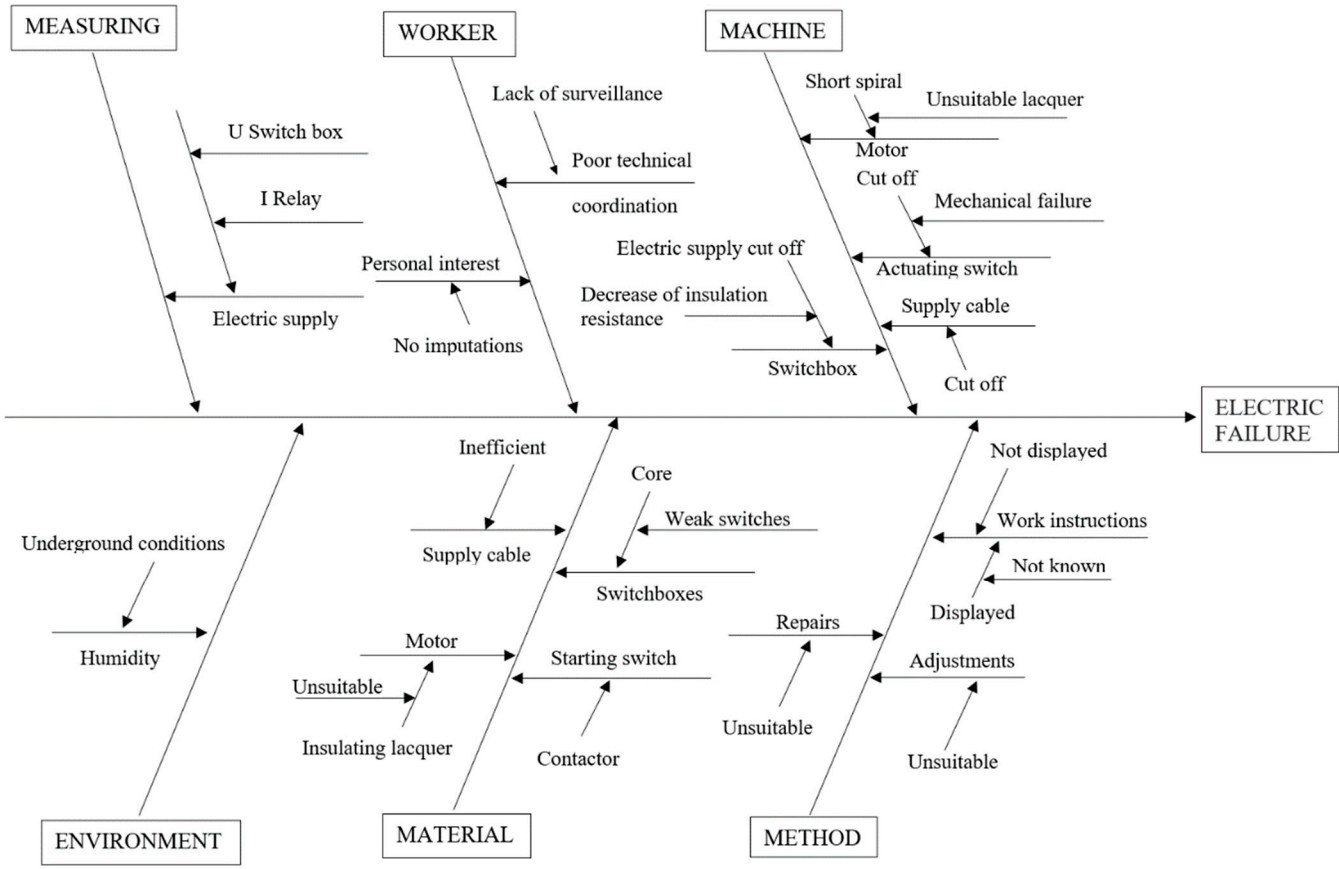

**Figure 5.** Ishikawa diagram (cause–effect) for electric failure.

The diagram in Figure 6 highlights that the interruptions in operation due to the combustion of the electric motor have the following causes: operating conditions in humidity and dust; operating personnel who are negligent, unresponsive, or poorly trained; the conveyer's insufficient transport capacity; the measurement and monitoring of the electrical parameters are deficient; and the insulating varnish is inadequate.

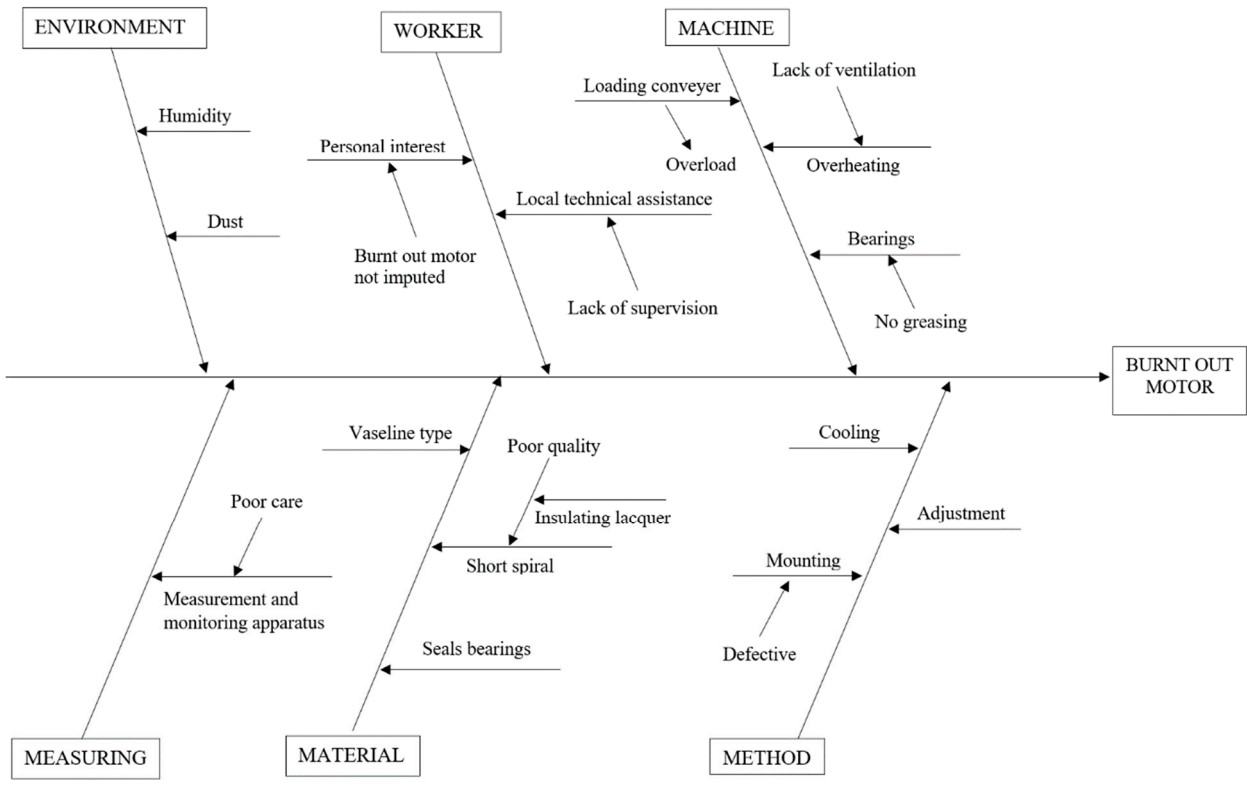

**Figure 6.** Ishikawa diagram (cause–effect) for burnt-out motor failure.

### 4.2. Possibilities of Streamlining TR-7A Conveyer Maintenance

The data obtained for the scraper conveyer TR-7A, regarding the number of failures, their frequency, and their remedial times (Table 2), were collected from one of Jiu Valley's mining exploitation sites, over a period of two years. The main defect found was in the conveyor chain (21.28%), for which the reliability and maintainability analysis presented below were carried out.

**Table 2.** Centralizer of the failures and repair times for the TR-7A conveyer.

| Part | Number of Failures | Failure Frequency $f_c$ (%) | Repair Time (min) | Repair Time Share $p_r$ (%) | MTR (min) |
|---|---|---|---|---|---|
| Chain | 10 | 21.28 | 530 | 16.56 | 53 |
| Failure of coupling | 9 | 19.15 | 580 | 18.13 | 6.44 |
| Electric failure | 7 | 14.89 | 290 | 9.06 | 41.43 |
| Born out motor | 6 | 12.77 | 390 | 12.19 | 65 |
| Switch PVI | 6 | 12.77 | 380 | 11.88 | 63.33 |
| Chain lifter | 4 | 8.51 | 420 | 13.12 | 105 |
| Gear assembly | 2 | 4.25 | 240 | 7.50 | 120 |
| Return drum | 2 | 4.25 | 230 | 7.19 | 115 |
| Drive drum | 1 | 2.13 | 140 | 4.37 | 140 |
| TOTAL | 47 | 100 | 3200 | 100 | |

The operating times between two faults and the repair times for the conveyor chain are specified in Table 3.

**Table 3.** Values of operating time between failures and time to repair.

| Time | Value [1] | | | | | | | | | |
|---|---|---|---|---|---|---|---|---|---|---|
| Operating times between two failures (hours) | 385 | 126 | 343 | 175 | 28 | 322 | 378 | 357 | 756 | 336 |
| Repair times (minutes) | 50 | 60 | 55 | 60 | 55 | 65 | 65 | 45 | 35 | 40 |

[1] values collected from Jiu Valley coal basin.

The data thus obtained have been analyzed using Weibull++ soft (ReliaSoft 2022, produced by ReliaSoft Company, Tucson, AZ, USA) with a view to obtaining the indicators and parameters of reliability and maintainability [37].

The stages followed in the analysis of reliability and maintainability are:

1.  Establishing the acceptance ranking of the distribution law suitable for the respective fault—in this example, it is the conveyer chain (Table 4)—and the remedy times, respectively (Table 5), the individual study of each repartition law no longer being necessary. Depending on the practical experience of the user, the first or one of the first laws of this classification can be used. In the present case, we chose the first laws (Tables 6 and 7).
2.  Determination of the reliability (Table 6) and maintainability (Table 7) indicators of the adopted law;

**Table 4.** Classification of the distribution law acceptance for the chain failure.

| Current Results Matrix Matrix Order: Distribution [1] | Ranking | LKV | BIC | AIC |
|---|---|---|---|---|
| Normal | 1 | −66.45 | 137.5 | 136.9 |
| 2P-Weibull | 2 | −66.48 | 137.6 | 137 |
| Gamma | 3 | −66.61 | 137.8 | 137.2 |
| 1P-Exponential | 3 | −67.75 | 137.8 | 137.5 |
| 2P-Exponential | 4 | −66.81 | 138.2 | 137.6 |
| 3P-Weibull | 5 | −66.34 | 139.6 | 138.7 |
| Loglogistic | 5 | −67.47 | 139.5 | 138.9 |
| Lognormal | 6 | −67.81 | 140.2 | 139.6 |
| Gumbel | 7 | −70.19 | 145 | 144.4 |
| G-Gamma | 8 | $-1 \times 10^{99}$ | $2 \times 10^{99}$ | $2 \times 10^{99}$ |

[1] using Weibull++ soft.

**Table 5.** Classification of the distribution law acceptance for the chain repair.

| Current Results Matrix Matrix Order: Distribution [1] | Ranking | LKV | BIC | AIC |
|---|---|---|---|---|
| 2P-Weibull | 1 | −37 | 78.51 | 77.91 |
| Normal | 2 | −37.1 | 78.86 | 78.25 |
| Logistic | 3 | −37.4 | 79.49 | 78.89 |
| G-Gamma | 3 | −36.4 | 79.64 | 78.74 |
| Gamma | 4 | −37.5 | 79.68 | 79.08 |
| Lognormal | 5 | −37.6 | 79.87 | 79.27 |
| Loglogistic | 6 | −37.9 | 80.39 | 79.78 |
| 3P-Weibull | 7 | −37.2 | 81.3 | 80.39 |
| 2P-Exponential | 8 | −39 | 82.6 | 81.99 |
| 1P-Exponential | 9 | −50.4 | 103 | 102.7 |

[1] using Weibull++ soft.

**Table 6.** Adopted exponential repartition law parameters for the chain failure.

| Results Reports [1] | |
| --- | --- |
| **Report Type** | **Weibull++ Results** |
| **User Info** | |
| Name | Vlad Alexandru Florea |
| Company | University of Petrosani |
| Date | 06.04.2022 |
| **Parameters** | |
| Distribution | Normal 2P |
| Analysis | RRX |
| CB Method | FM |
| Ranking | MED |
| Mean (hr) | 320.600028 |
| Std (hr) | 196.819109 |
| LK Value | −66.449307 |
| Rho | 0.924127 |
| Fail\Susp | 10\0 |
| **LOCAL VAR/COV MATRIX** | |
| Var-Mu = 3873.776177 | CoVar = 0.000671 |
| CoVar = 0.000671 | Var-Sigma = 2330.451737 |

[1] using Weibull++ soft.

**Table 7.** Adopted exponential repartition law parameters for the chain repair.

| Results Reports [1] | |
| --- | --- |
| **Report Type** | **Weibull++ Results** |
| **User Info** | |
| Name | Vlad Alexandru Florea |
| Company | University of Petrosani |
| Date | 06.04.2022 |
| **Parameters** | |
| Distribution | Weibull 2P |
| Analysis | RRX |
| CB Method | FM |
| Ranking | MED |
| Beta | 5.411076 |
| Eta (hr) | 57.303281 |
| LK Value | −36.954764 |
| Rho | 0.986964 |
| Fail\Susp | 10\0 |
| **LOCAL VAR/COV MATRIX** | |
| Var-Beta = 2.130158 | CV Eta Beta = 0.418039 |
| CV Eta Beta = 0.418039 | Var-Eta = 12.663176 |

[1] using Weibull++ soft.

### 4.3. The Analysis of the Fundamental Reliability and Maintainability Indicators

Figures 7–11 show the functions: of failure probability (Figure 7), unreliability (Figure 8), density of probability (Figure 9), failure rate (Figure 10), and repair rate, respectively (Figure 11).

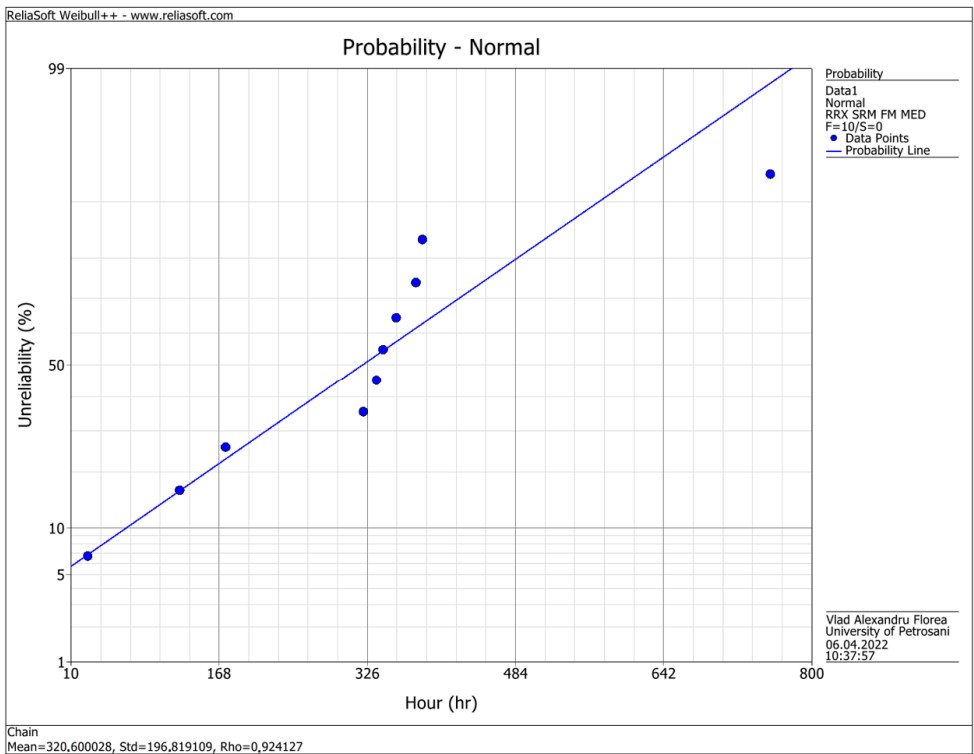

**Figure 7.** Probability function for the chain.

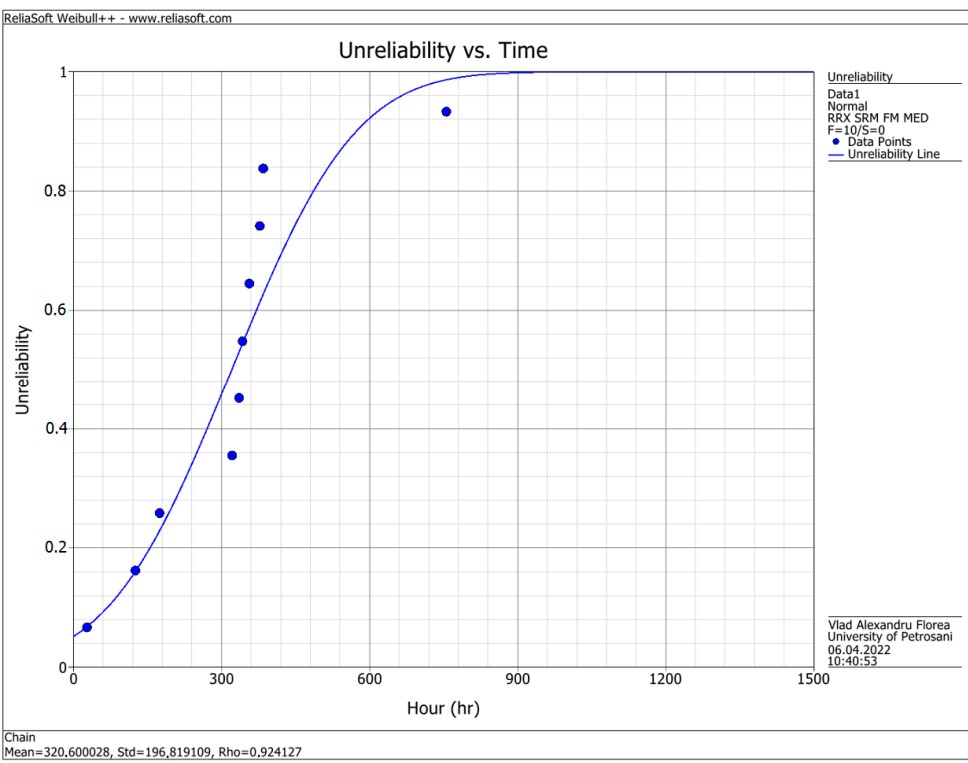

**Figure 8.** Unreliability function for the chain.

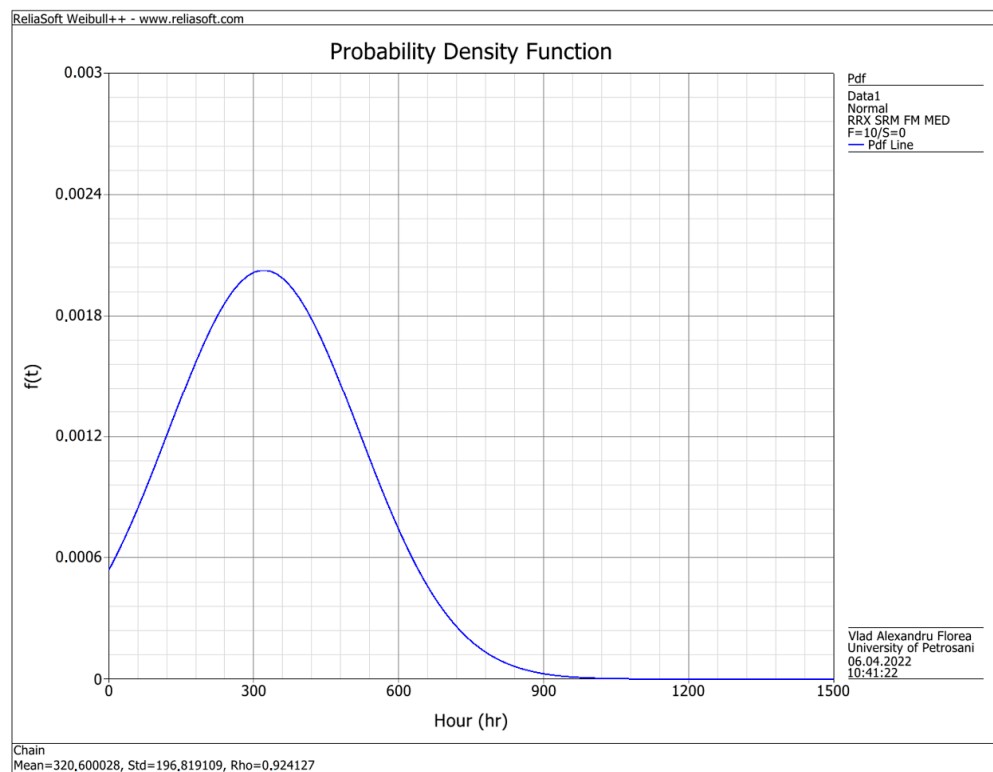

**Figure 9.** Probability density function for the chain.

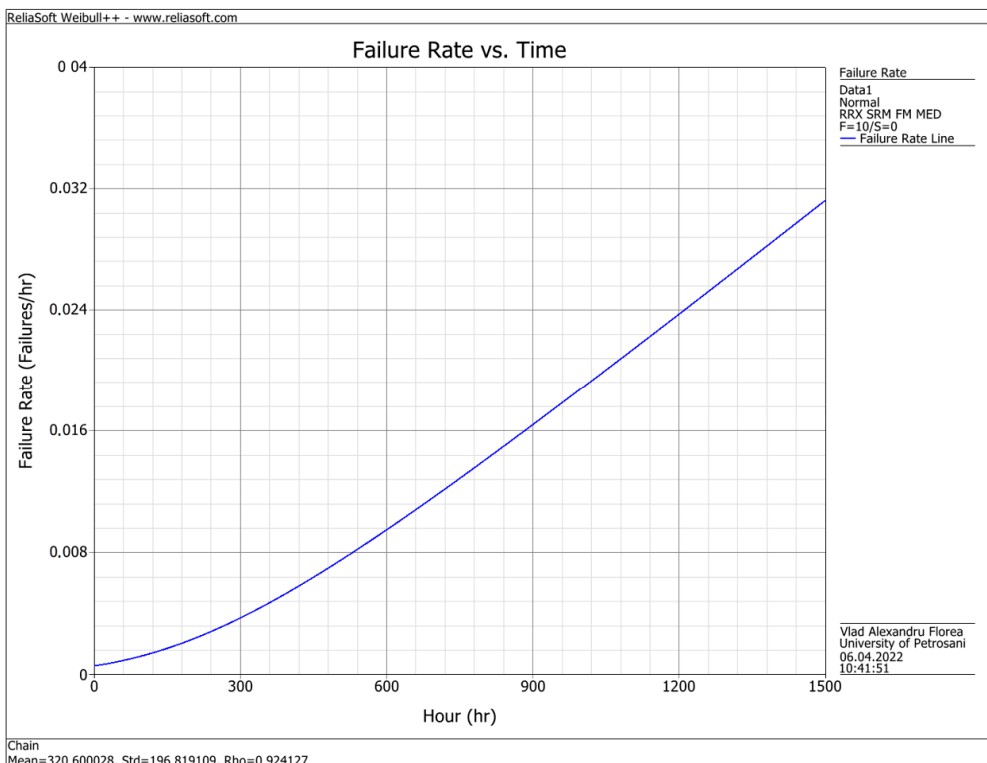

**Figure 10.** Failure rate for the chain.

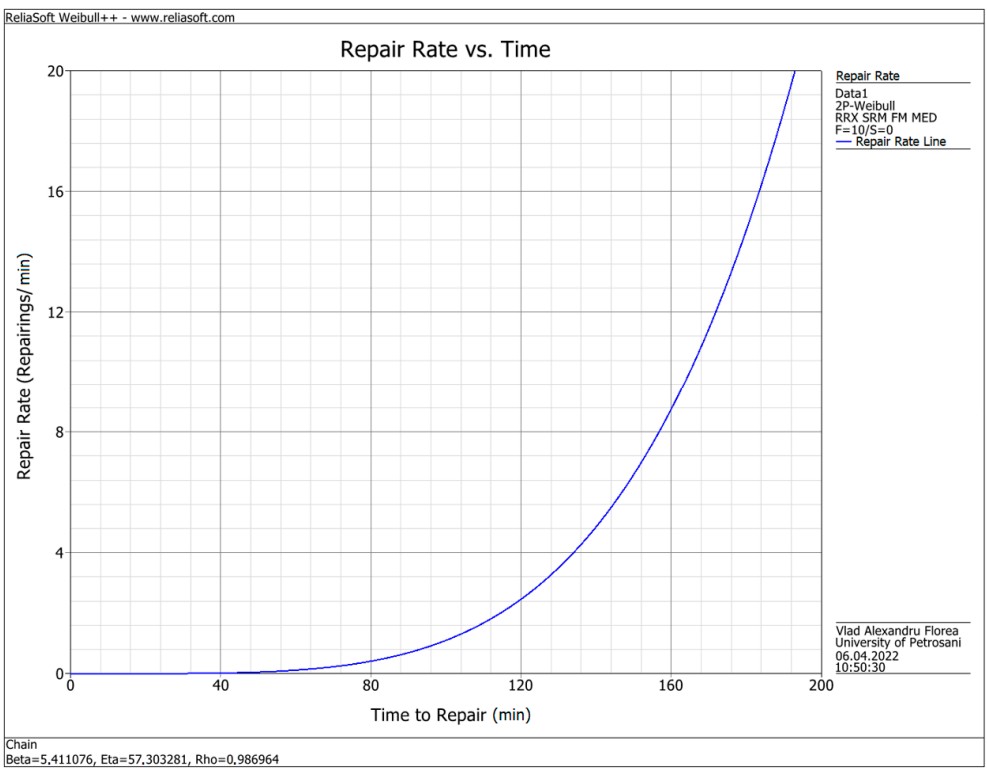

**Figure 11.** Repair rate for the chain.

The great number of the results obtained allowed graphs to be developed regarding the chain reliability evolution in time (Figures 12 and 13).

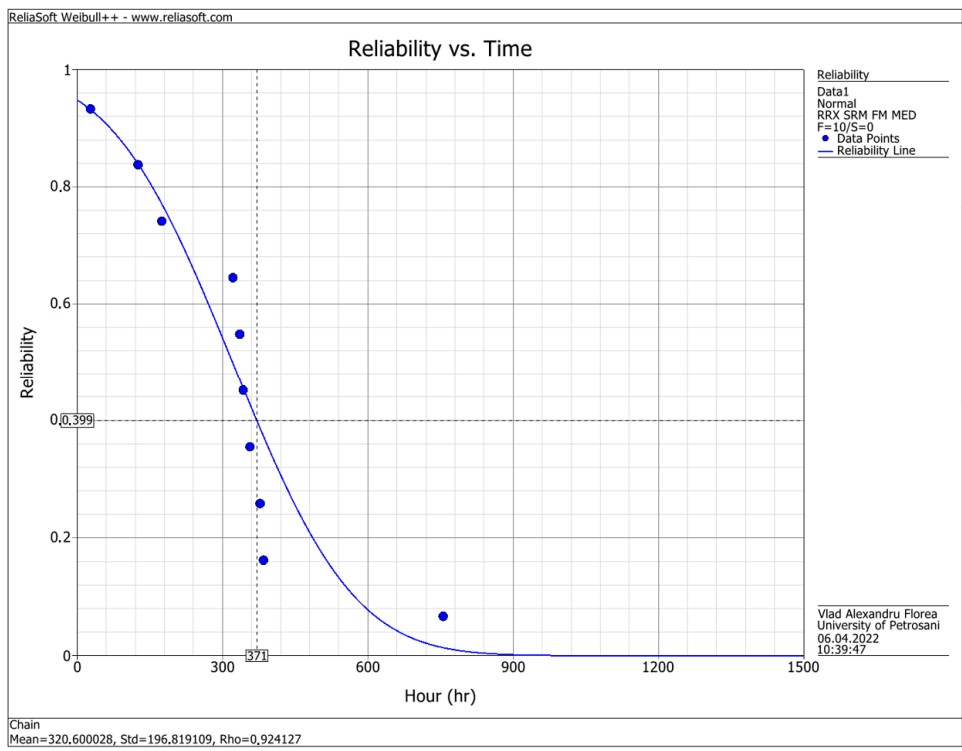

**Figure 12.** Chain reliability values, for a 39.9% confidence level.

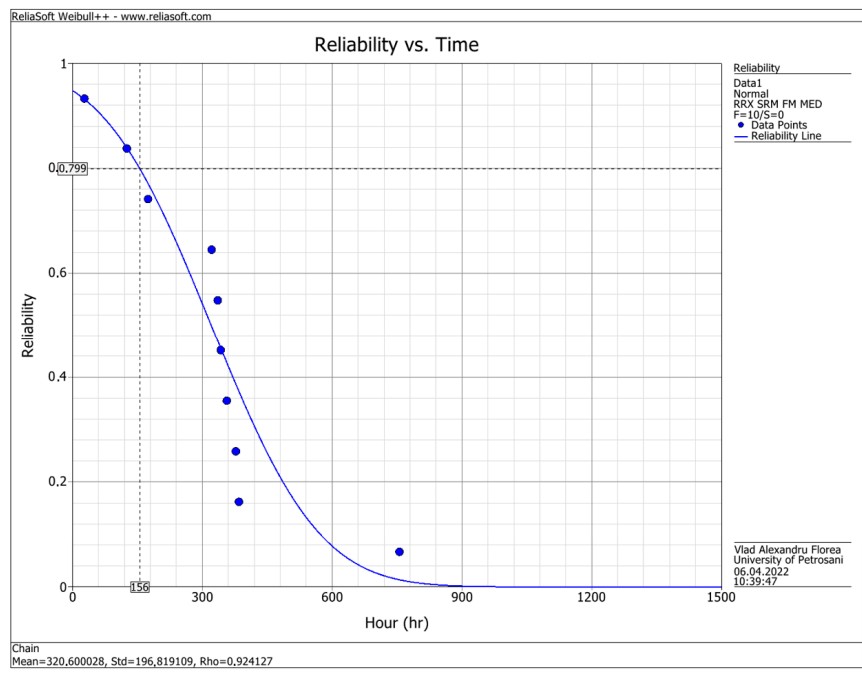

**Figure 13.** Chain reliability values, for a 79.9% confidence level.

From these graphs, we see low values of the conveyer chain reliability result, so that the probability for it not to fail after 371 h of operation in effect is as low as 40% (Figure 12). This shows that for a 40% confidence level (a 60% risk margin being very high), we should expect the replacement of the chain after approximately 15 days of operation (the conveyer operates 5 h a day for 5 days a week).

If an 80% reliability is imposed (Figure 13), which is a value imposed by the beneficiaries of the underground mining equipment, only 156 h of operation will result in no failure, which means that after approximately 7 days of operation in effect, it will be necessary to replace the chain.

From the analysis of the maintainability of the conveyer chain, it is noticed that in order to replace the chain in approximately 47 min, the maintainability is only 30% (Figure 14), with an increase of approximately one hour for a maintainability of 80% (Figure 15).

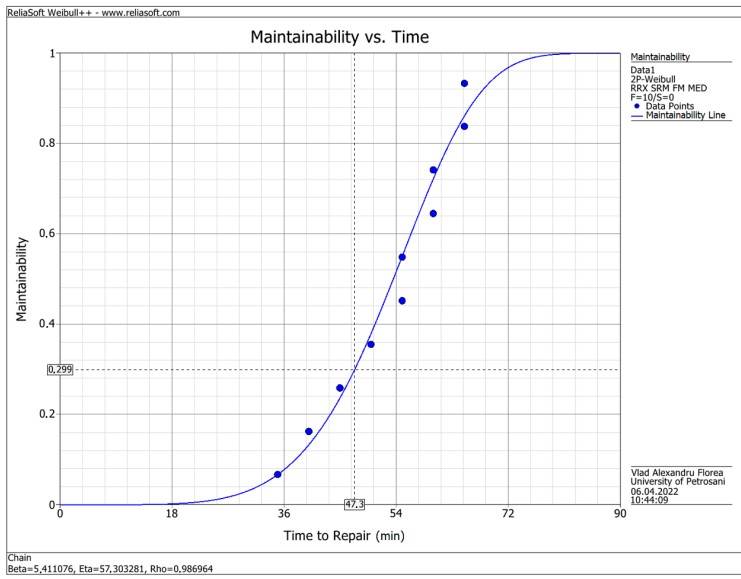

**Figure 14.** Chain maintainability values, for a 29.9% confidence level.

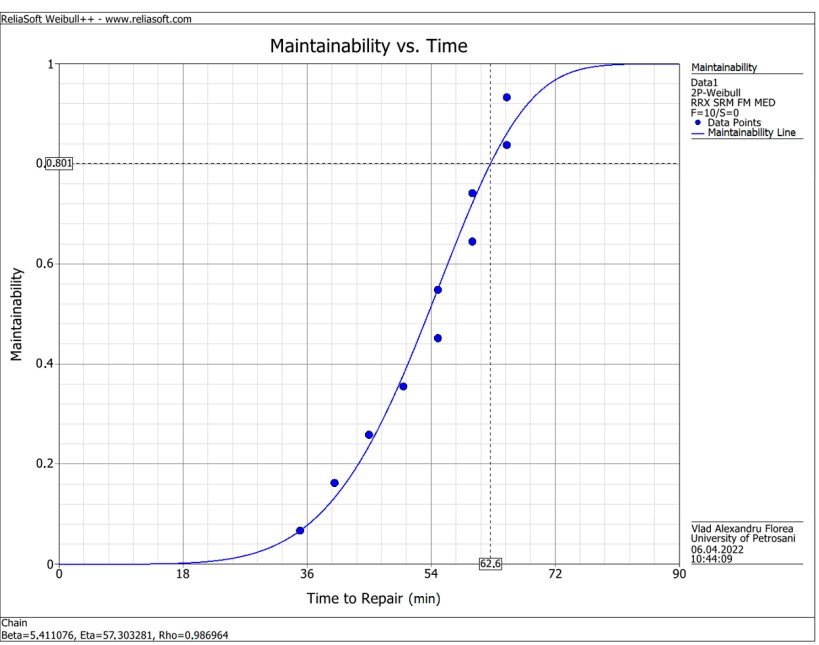

**Figure 15.** Chain maintainability values, for an 80.1% confidence level.

## 5. Conclusions

Maintenance means keeping the workplace, structures, equipment, and machines in working order and in complete safety, as well as ensuring that their condition does not deteriorate. Maintenance is a process that touches all areas of occupational safety and health, covers every job, and concerns personnel at all levels, not only maintenance workers.

The correct choice of the machinery type with the technical characteristic correlated with the exploitation conditions is a prime necessary condition to be fulfilled with a view to obtaining an optimum availability.

It would fall upon the mines and mining machine, machinery, and equipment building plants, as a permanent obligation, to pay attention to the following aspects: following and analyzing the modifications of technical–operational parameters, discovering the causes that lead to the lessening of certain characteristics, and pinpointing the causes of accidental failures.

The graphical–logical, causal, and suggestive analysese, performed in this paper, using Ishikawa diagrams, allow efficient and economical action on the technological parameters of the machinery's exploitation process.

By means of the Ishikawa diagrams, we highlighted the causes leading to the failure of the TR-7A conveyer components (with the higher share of faults), as resulted from the analysis of the machinery's reliability and maintainability.

The results obtained regarding the reliability of the chain of the TR-7A scraper conveyer have shown the necessity of improvement of the performances of the conveyer chain, in order to increase its operating life.

The physical–mechanical properties, wear resistance, and granulometry of the rocks influence the intensity of the wear phenomenon that they cause on the components of the mining equipment, including the conveyor. It was found that, in mining works, the rocks have an inhomogeneous structure, which means that the initial assessment of their influence on the wear and reliability of the conveyer chain is necessary through laboratory studies (on rock samples).

During the technical inspection of the chain, areas with deformations and reductions in its dimensions can be found. The wear of the chain leads to its frequent breakage and the need to replace some sectors in the area where the breakage occurred.

Rigorously executed maintenance activities, with skilled staff, can lead to a shortening of the timelapse required for not only the chain replacement, but also for other subassemblies that undergo frequent failures, as well as to lower exploitation expenses.

The obtained results from the analysis of the maintainability of the four components with the highest number of defects allowed for the improvement and efficiency of the system of technical revisions and preventive-planned repairs used in the case of the TR-7A conveyer.

Chain breakage is due to both the material and its manufacturing technology. Taking into account the fact that the elements of the chain form frictional couplings with the inner surface of the conveyor chutes, an increase in wear resistance can be achieved by increasing the content of chromium, manganese, or tungsten in the composition of the materials of both components, which can lead to a considerable increase in manufacturing costs.

The maintenance process is at the heart of good work practices in safety and health conditions. While the specific details of the activities carried out by the maintenance staff vary at the level of industrial sectors, depending on the work equipment used, there are a series of principles common to the actual maintenance in all workplaces. These include the need for maintenance to start with proper planning, covering health and safety issues, and following a structured approach based on risk assessment, with clear roles, guidelines, and responsibilities for field workers, appropriate training and equipment, and periodic checks to ensure the good performance of the activity and the fact that no new risks are created. It is essential that maintenance be seen as a process that must be managed systematically and not as a simple single obligation. Workplaces need an integrated approach to maintenance, based on a risk assessment that takes into account the safety and health aspects at every stage of the maintenance process, and directly involves workers in the maintenance management process.

**Author Contributions:** Literature review and analysis, V.A.F. and M.T.; methodology, V.A.F. and R.-B.I.; writing, M.T. and V.A.F.; experiments, R.-B.I.; results analysis, V.A.F. and M.T. All authors have read and agreed to the published version of the manuscript.

**Funding:** This research received no external funding.

**Institutional Review Board Statement:** Not applicable.

**Informed Consent Statement:** Not applicable.

**Data Availability Statement:** Not applicable.

**Conflicts of Interest:** The authors declare no conflict of interest.

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
