# Peer review of "Assessment Possibilities of the Quality of Mining Equipment and of the Parts Subjected to Intense Wear"

_applsci, doi:10.3390/app13063740_

Round 1

Reviewer 1 Report

The presented study is relevant for mining operations. Equipment downtime has a huge impact on the productivity of mines and therefore reduces their economic efficiency. One of the important conditions for the normal operation of mining equipment is their proper maintenance and the use of quality materials in their manufacture. The use of diagrams undoubtedly makes it possible to evaluate the structural influence of various parameters on the reliability of equipment. The authors have done a lot of work in the field of observation and diagnostics of equipment, however, a number of comments can be made:

1. The authors present well-known methods for calculating the reliability of equipment.

2. The authors describe the main causes of equipment failure, but do not indicate the share of each of the failures in the total number of failures. It is recommended to supplement the text of the article with statistical data on the share of each failure.

3. It is also not clear from the article which failure has the strongest impact on equipment downtime, that is, for example, which part replacement requires the most time.

4. The authors singled out a conveyor chain for their analysis, while not substantiating why the chain was chosen.

5. Information is not provided on the condition under which it is necessary to replace the chain, which parameter must be monitored during technical inspection.

6. In the article, the main cause of failure is equipment wear, but when working on different rocks, the wear rate can change, how can this parameter be taken into account in the reliability calculation model?

7. The conclusions make recommendations for the use of more wear-resistant materials, but how will the use of new materials affect the cost of mining?

The article is recommended for publication after the correction of comments.

Reviewer 2 Report

 In this paper, there are many fault types of mining equipment components, and few quality assessment methods for components. Take the model TR-7A scraper conveyor as an example, and use Ishikawa diagram to analyze the causes of the fault logic of causality and implication. The analysis of faults and repair times of the conveyor chain parts is obtained to maintain and extend the continuous running time of the conveyor, aiming to improve the maintenance efficiency of TR-7A scraper conveyor and improve its service life.

 Section 2 mentioned the four faults of transport machine, respectively is broken chain fault, coupling fault, electrical fault and motor burning fault, but the fourth article only analyzes the broken chain fault, since the no longer need to analyze the rest of the fault separately, figure 4 to figure 6 to explain the cause of the other three faults is necessary. And the article does not have the relevant explanation of the fault ishikawa diagram.

 Table 3 to Table 6 in section 4 of the article do not explain the content in the table, which makes it less related to the failure function in the third section of the article. We hope that the author can supplement the relevant content.

 Whether the fitting function of the probability function of chain failure in Figure 7 is accurate and whether higher order functions are required for fitting. I hope the author can make relevant explanations.

 Conclusion There is no rigorous explanation for the summary of the previous data, especially for the replacement cycle of chain faulty parts.

Round 2

Reviewer 2 Report

The author has revised the paper well. I agree to accept the revised thesis.